# Conformational Effects of Pt-Shells on Nanostructures and Corresponding Oxygen Reduction Reaction Activity of Au-Cluster-Decorated NiO*_x_*@Pt Nanocatalysts

**DOI:** 10.3390/nano9071003

**Published:** 2019-07-11

**Authors:** Dinesh Bhalothia, Yu-Jui Fan, Yen-Chun Lai, Ya-Tang Yang, Yaw-Wen Yang, Chih-Hao Lee, Tsan-Yao Chen

**Affiliations:** 1Institute of Electronics Engineering, National Tsing Hua University, Hsinchu 30013, Taiwan; 2Department of Engineering and System Science, National Tsing Hua University, Hsinchu 30013, Taiwan; 3School of Biomedical Engineering, Taipei Medical University, Taipei 11031, Taiwan; 4National Synchrotron Radiation Research Center, Hsinchu 30007, Taiwan; 5Institute of Nuclear Engineering and Science, National Tsing Hua University, Hsinchu 30013, Taiwan; 6Hierarchical Green-Energy Materials (Hi-GEM) Research Center, National Cheng Kung University, Tainan 70101, Taiwan; 7Higher Education Sprout Project, Competitive Research Team, National Tsing Hua University, Hsinchu 30013, Taiwan

**Keywords:** oxygen reduction reaction, nanocatalysts, carbon nanotube, wet-chemical reduction method, Au-clusters, mass activity

## Abstract

Herein, ternary metallic nanocatalysts (NCs) consisting of Au clusters decorated with a Pt shell and a Ni oxide core underneath (called NPA) on carbon nanotube (CNT) support were synthesized by combining adsorption, precipitation, and chemical reduction methods. By a retrospective investigation of the physical structure and electrochemical results, we elucidated the effects of Pt/Ni ratios (0.4 and 1.0) and Au contents (2 and 9 wt.%) on the nanostructure and corresponding oxygen reduction reaction (ORR) activity of the NPA NCs. We found that the ORR activity of NPA NCs was mainly dominated by the Pt-shell thickness which regulated the depth and size of the surface decorated with Au clusters. In the optimal case, NPA-1004006 (with a Pt/Ni of 0.4 and Au of ~2 wt.%) showed a kinetic current (*J*_K_) of 75.02 mA cm^−2^ which was nearly 17-times better than that (4.37 mA cm^−2^) of the commercial Johnson Matthey-Pt/C (20 wt.% Pt) catalyst at 0.85 V vs. the reference hydrogen electrode. Such a high *J_K_* value resulted in substantial improvements in both the specific activity (by ~53-fold) and mass activity (by nearly 10-fold) in the same benchmark target. Those scenarios rationalize that ORR activity can be substantially improved by a syngeneic effect at heterogeneous interfaces among nanometer-sized NiO*_x_*, Pt, and Au clusters on the NC surface.

## 1. Introduction

Fuel cells are expected to be commercially feasible to moderate deficiencies in natural energy resources without increasing the carbon footprint [1,2,3,4]. In spite of many fascinating features like noise-free operation, substantial reductions in pollution, and better efficiencies, the commercial viability of fuel cells is hindered by the substantially high energy barrier of oxygen reduction reactions (ORR) at the cathode [5,6]. To reduce the energy barrier to ORR, platinum (Pt)-based heterogeneous catalysts seem to be the most effective material [7,8,9]. Due to the unaffordable costs and low storage potential of Pt, finding alternative materials for nanocatalysts (NCs) with comparable efficiencies to Pt is an inevitable step to bringing fuel cells into the market. Meanwhile, lower overpotential losses, long-term durability, pH working conditions, non-toxicity, and earth-abundant elements are fundamental physical and economic requirements for usable material combinations. Despite many efforts so far expended on the development of fuel cells, especially over the past two decades, many hurdles have yet to be overcome. Several Pt-alloys [10,11,12], including 3D-transition metals (Co, Ni, Cu, Fe, etc.) together with core shell nanostructures [13,14,15], bimetallic nanodendrites [16], nanowires [17], nano-onions, etc. [18], have been intensively studied in recent decades. In addition, great efforts have been geared towards size [19], shape [20], and composition [21] controls of Pt-based NCs to overcome the aforementioned challenges for preparing highly active ORR catalysts. Those studies laid a strong foundation to further fine-tune the electronic and chemical properties of NCs to extend ORR performances. Promising and efficient techniques, however, are still far away from attaining commercial standards.

Achieving a reconcilable balance between catalytic activity and noble-metal dosages when developing NCs for ORRs is still a challenging task. Core-shell structured heterogeneous NCs with a transition metal (e.g., Co, Ni, Zn, Ru, Fe, and Sn) in the core and a Pt shell seem to be the most effective design in terms of cost considerations and catalytic activities. In such configurations, the core crystal injects electrons (or forms a negative field) to the shell crystal via a combination of three major effects: A bifunctional mechanism [22] (using a variety of adsorption species), ligand effects (electron localization because of the electronegativity gap between two atoms) [23,24,25], and the lattice strain (differences in atomic arrangements between intraparticle domains) [26,27,28]. Moreover, such elements, owing to low-energy pathways, provide the opportunity for allocation and recombination kinetics of radicals (i.e., O*, OH*, and H*) in H_2_O and reduce durations of intermediate steps on NC surfaces. Meanwhile, the Pt-shell protects transition metals in the core from corrosive conditions at fuel cell cathodes.

The presence of Au in heterogeneous NCs offers electronic, geometric, and compositional effects to tune catalytically active sites that were found to be effective for ORR [29,30]. We further improved the ORR activities of such NCs via decorating strong electronegative atomic-scale Au clusters at the interface and on the surface of Pt-stacked transition metal nanocrystallites. Au clusters not only recover surface defects but also form indirect heterojunctions to the core crystal and localize valence electrons from neighboring atoms using strong electronegative forces. Meanwhile, Pt forms an unconformable shell, which protects the core crystal from corrosion and shares ORR pathways, including O_2_ splitting and relocation kinetics of O-atoms, and thus avoids highly energetic intermediates and their associated kinetic penalties. Our previous work demonstrated a facile approach to trigger ORR activity via Pt-decorated core-shell structures [31,32,33,34,35]. Those ternary metallic NCs consisting of lower dosages of Pt showed distinct activity towards ORR facilitation but with reduced fabrication costs.

This study implemented an innovative sequence and time-controlled wet-chemical reduction method to synthesize Au cluster-decorated Ni_core_-Pt_shell_ NCs. The inner structure (heterogeneous intra-/interparticle interfaces and lattice strain) and surface coverage of such NCs were altered via changing the dosages of Au and Pt. In this event, a series of carbon nanotube (CNT)-supported ternary metallic NCs comprising a Ni/NiO_*x*_ core and an Au cluster-decorated Pt-shell (called NPA) were synthesized with variable shell thicknesses (with Pt/Ni ratios of 0.4 and 1.0) and Au contents (~2 and 9 wt.%) on the surface. Such synthesized NCs with unique multiphase cluster-in-cluster interfaces and surface modifications preserved improved catalytic activities towards ORRs in an alkaline environment (0.1 M KOH). Of greatest relevance, the mass activity (MA) and kinetic current density (*J_k_*) of NPA-1004006 (with a Pt/Ni ratio of 0.4 and Au of ~2 wt.%) were improved by 10.36- and 17.16-fold, respectively, compared to those of commercial the Johnson Matthey-Pt/C (20 wt.% Pt) catalyst. At the same time, multiple metallic interfaces contributed to the lower electrochemical surface area (ECSA) with a shift of the Pt oxide reduction peak to more-positive potentials which thus improved the specific activity (SA) by 53.21-fold that of the commercial Johnson Matthey-Pt/C catalyst. Herein, our findings present a proper strategy for the design of heterogeneous NCs by managing their local structure through controlling the surface and inner configurations. Systematic interpretations of the experimental results are given in latter sections.

## 2. Experimental

### 2.1. Synthesis Methodology and Materials for Preparing Ni Core-Pt Shell (Ni@Pt) NCs

Ni@Pt NCs were synthesized using a sequential wet-chemical reduction method [32]. Scheme 1 reveals the reaction steps of the synthesis methodology. Prior to NC synthesis, surface functionalization of catalyst support (multi-walled CNT (MWCNT), Cnano Technology Ltd., Beijing, China) was carried out via acid-treatment in 4.0 M H_2_SO_4_ at 80 °C for 6 h. In this way, the attachment of metallic crystals onto the CNT surface was strengthened. First, 500 mg of MWCNTs (5 wt.% solution in ethylene glycol (EG)) was added as catalyst support in 1.28 g of an aqueous solution, which contained 0.1 M nickel (II) chloride hexahydrate (NiCl_2_**·**6H_2_O, Showa Chemical Co. Ltd., Tokyo, Japan), and then this was stirred at 200 rpm for 6 h. The mixture (Ni^2+^ adsorbed CNT; CNT-Ni^ads^) contained 0.128 mmoles (7.5 mg) of Ni metal ions in a metal loading of Ni/CNT of 30 wt.%. After stirring, 5 mL of a water solution with 0.0386 g NaBH_4_ (99%, Sigma-Aldrich, St. Louis, MO, USA) was added (step 2) to samples prepared in the first step and stirred at 200 rpm for 10 s. After that, metastable Ni metal nanoparticles (NPs) were formed (Ni/CNT sample). In step 3, 1.28 g of a Pt precursor solution, 0.128 mmoles Pt metal ions (i.e., 0.1 M), was added to the Ni/CNT sample, and then a thin layer of Pt crystals on the surface of nickel NPs (namely Ni@Pt-1010) was formed. In this step, Pt ions could be reduced by an excessive amount of NaBH_4_ added in step 2 and deposited on the Ni surfaces. According to different Pt/Ni molar ratios, Pt shells with different thicknesses were obtained. The Pt precursor solution was prepared by diluting ∼1.0 g H_2_PtCl_6_·6H_2_O (99%, Sigma-Aldrich Co., Burlington, MA, USA) to 18.36 g with distilled water. In the remainder of this article, Ni@Pt NCs with Pt/Ni atomic ratios of 1.0 and 0.4 are called Ni@Pt-1010 and Ni@Pt-1004, respectively.

### 2.2. Synthesis of Atomic Au Cluster-Decorated Ni Core-Pt Shell (NPA) NCs

A precursor solution of Au was prepared by diluting ∼13.3 mg HAuCl_4_·3H_2_O (99.0%, Sigma-Aldrich Co., Burlington, MA, USA) to 500 mg with distilled water. After 20 min, once the reaction was complete in step 3, an appropriate amount of the prepared Au solution was added to a solution to make Au/Pt atomic ratios of 0.06 and 0.2. During this stage, Au clusters formed and were intercalated in the Pt shell region through a galvanic replacement reaction of Au^3+^ with Pt, and then with the reducing agent, NaBH_4_, these metal ions decreased. Finally, products of atomic Au cluster-decorated Ni@Pt (NPA) NCs were obtained. The resulting precipitate was washed several times with acetone, centrifuged, and then dried at 70 °C. In the remainder of this article, Au cluster-deposited Ni@Pt-1004 NCs are called NPA-1004006 and NPA-100402 for 2 and 9 wt.% Au loadings, respectively. Furthermore, Au cluster-deposited Ni@Pt-1010 NCs were named NPA-1010006 and NPA-101002 for 2 and 9 wt.% Au loadings, respectively.

### 2.3. Physical Characterizations of NPA NCs

Physical properties of the prepared NCs were determined by cross-referencing results of microscopic and X-ray spectroscopic techniques. High-resolution transmission electron microscopic (HRTEM) characterizations were carried out at the electron microscopy center of National Sun Yat-Sen University (Kaohsiung, Taiwan). X-ray photoemission spectroscopy (XPS) of the experimental NCs was executed at beamlines BL-24A1 of the National Synchrotron Radiation Research Center (NSRRC) (Hsinchu, Taiwan). X-ray diffraction (XRD) patterns were collected at an incident X-ray wavelength of 1.5406 Å (8.04 keV) at Taiwan beamline of BL-12B2 in Spring-8 (Hyogo, Japan).

### 2.4. Preparation of the Electrode and the Method for the ORR Activity Experiment

Catalyst ink for the ORR experiment was made by dispersing 5.0 mg of catalyst powder in a solution consisting of 1.0 mL isopropanol and 50 μL Nafion-117 (99%, Sigma-Aldrich Co., Burlington, MA, USA). This mixture was subjected to ultrasonication for 30 min prior to the ORR test. To conduct the ORR test, 10.0 μL of catalyst ink was drop-cast and air-dried on a glossy carbon rotating disk electrode (RDE) (0.196 cm^2^ in area) as the working electrode. The Hg/HgCl_2_ (with the voltage calibrated to 0.242 V, to align with that of the reference hydrogen electrode (RHE)) electrode saturated in a KCl aqueous solution and a platinum wire were respectively used as the reference and counter electrodes. The ECSAs of the experimental catalysts were calculated by acquiring the columbic charge for reduction of the monolayer Pt oxide after integration and double-layer correction using the following equation:(1)$$ECSA=\frac{Q_{Pt}}{Q_{ref}\times m};$$
where *Q_ref_* is the charge required for reduction of the monolayer oxide from the bright Pt surface (i.e., 0.405 mC cm^−2^), m is the metal loading, and *Q_Pt_* is the charge required for oxygen desorption, as calculated by following equation:(2)$$Q_{Pt}=\frac{1}{\upsilon }\int \left(I-I_{d}\right)dE.$$

Here, υ is the scan rate for the cyclic voltammetric (CV) analysis, and integral parts refer to the area under the Pt oxide reduction peak on CV curves. The kinetic current density (*J_K_*) and number of electrons transferred in ORRs were calculated based on the following equations:(3)$$\frac{1}{J}=\frac{1}{J_{K}}+\frac{1}{J_{L}}=\frac{1}{J_{K}}+\frac{1}{B\omega^{0.5}}\;\mathrm{and}$$
(4)$$B=0.62nFC_{O_{2}}D_{O_{2}}{}^{\frac{2}{3}}\nu^{-\frac{1}{6}};$$
where *J*, *J_K_*, and *J_L_* are the experimentally measured, mass transport free kinetic, and diffusion-limited current densities, respectively. ω is the angular velocity of the electrode, n is the transferred electron number, F is the Faraday constant, *C_O_*_2_ is the bulk concentration of O_2_, *D_O_*_2_ is the diffusion coefficient, and *v* is the kinematic viscosity of the electrolyte. For each NC, the MA and SA were respectively obtained when *J_K_* was normalized to the Pt loading and ECSA. Details of the procedure for the ORR mass activity calculation are given in Appendix A.

### 2.5. Electrochemical Measurements

Electrochemical measurements were carried out at room temperature (25 ± 1 °C) using a potentiostat (CH Instruments Model 600B, CHI 600B; Hsinchu, Taiwan) equipped with a three-electrode. Cyclic voltammetry (CV) and linear sweep voltammetry (LSV) data were measured at voltage scan rates of 0.02 and 0.001 V s^−1^, potential ranges of 0.1~1.3 V (vs. the RHE.) and 0.4~1.1 V (vs. the RHE) in an aqueous alkaline electrolyte solution of 0.1 M KOH (pH 13). A rotation rate of 400~3600 rpm was used for LSV. N_2_ and O_2_ atmospheres were used for CV and LSV, respectively.

## 3. Results and Discussion

The particle shape, near-surface configurations, and crystal structure of the experimental NCs were determined using HRTEM analyses. As shown in Figure 1a, Ni@Pt-1004 NPs (Pt/Ni ratio = 0.4) had ordered atomic arrangements at (111) facets exposing the surface (denoted by red arrows). The presence of a clear twin boundary (denoted by a white line) suggested the formation of semi-coherent interfaces. The interplanar spacing of the (111) facet was determined to be 2.239 Å. This value is about 1.25% smaller than that of the Pt-CNT (2.267 Å) indicating the presence of compressive lattice strain in the shell region. With an average particle size of 2.64 nm in (111) facets (determined by an XRD analysis described in a later section), the surface-to-bulk ratio was estimated to be ~50% In this event, considering the ideal case of conformal deposition of Pt atoms by chemisorption and reduction, formation of an incomplete Pt shell over the Ni-core crystal was expected. In contrast, Ni@Pt-1010 NPs (Figure 1d) had grown into isotropic spheres, which comprised a complete Pt-shell and Ni core crystal underneath, and this was also confirmed by XPS analyses in a later section. Compared to that of Ni@Pt-1004 NPs, a higher extent of surface defects (denoted by yellow arrows) was observed in Ni@Pt-1010 NPs. The average particle size of Ni@Pt NCs was observed to be around 2~3 nm, which is consistent with XRD findings in a subsequent section.

Shown in Figure 1b, compared to that of Ni@Pt-1004 NPs, the diameter of NPA-1004006 (i.e., Ni@Pt-1004 NC decorated with 2.0 wt.% of Au atoms) nearly doubled by mixing with Au^3+^ ions for 2 min, followed by the addition of a reducing agent (NaBH_4_). NPA-1004006 particles grew in a disk-like shape (Figure 1b) with twin boundaries between the NPs. With a nearly identical coherent length to that of Ni@Pt-1004, the large particle was an agglomeration cluster with structural modulations at semi-coherent interfaces between NPs by the galvanic replacement of Au^3+^ to core (Pt/Ni) metals simultaneously with the reduction of residual Pt^4+^, Ni^2+^, and Au^3+^ ions by the reducing agents. Such a hypothesis was confirmed by XRD and XPS analyses in later sections. With an incomplete Pt shell structure, certain parts of the NiO_*x*_ core were exposed to the liquid environment. In this event, a high content of Ni atoms participated in the galvanic replacement reaction by interacting with Au^3+^ ions. Therefore, Au atoms tended to form metallic clusters between the NPs. As shown in Figure 1c, with a further increase of Au loading to 9 wt.% (NPA-100402), discrete local domains were formed with lattice fringes pointing in different directions (denoted by green arrows). Such a characteristic further revealed severe galvanic replacement of the NiO_*x*_ core followed by agglomeration of Ni@Pt NPs by reduction and deposition of residual metal ions in their interfaces. In this event, because of sufficiently high Au loading, the majority of Au atoms were deposited on the top of and between the NPs. Compared to that of Ni@Pt-1004, restructuring of Ni@Pt-1010 by Au^3+^ was insignificant. As shown in Figure 1e,f, particles tended to grow in a core-shell structure with highly ordered atomic arrangements on the surface with increasing Au contents. Such a characteristic was consistently proven by an XRD analysis, which showed that the coherent length of the Pt crystal (111) facet had increased from 2.72 to 3.34 ± 0.1 nm by adding 2.0 wt.% of Au atoms (namely NPA-1010006). Compared to that of NPA-1010006, further increasing Au atoms to 9 wt.% reduced the coherent length to 3.03 ± 0.1 nm, which could be attributed to restructuring between Pt crystals and Au^3+^ ions followed by formation of Au crystals in the shell region (shown by the presence of a shoulder on the left hand side of the Pt (111) peak). EDX results of NPA-1004006 NC has been shown in Appendix A.

Lattice strain, the coherent length (*D_avg_*), and the crystallinity of pristine NCs were revealed by XRD analyses. Figure 2 compares XRD patterns of the experimental NCs. Values of the *D_avg_* of experimental NCs were calculated from the XRD peak broadening of (111) facets using the Scherrer equation, and their corresponding structural parameters are summarized in Table 1. As indicated in Figure 2, peaks *X*1 and *X*2 centered at 40.2° and 46.4° respectively refer to diffraction signals from (111) and (200) facets of metallic-phase Pt nanocrystals in the NCs. In an XRD pattern, the peak width denotes the relative dimension of long-range ordering in specific facets (i.e., the coherent length, *D_avg_*) and the ratio of peak intensities between the (111) and (200) facets (i.e., h (111)/h (200)), which refers to the extent of preferential crystal growth for samples under investigation. For Ni@Pt NCs (Appendix A), the *D_avg_* increased by 0.1 nm, which could be attributed to the formation of thin layers of Pt in the shell region over the Ni core crystal with Pt/Ni ratios of 0.4 (Ni@Pt-1004) and 1.0 (Ni@Pt-1010) (Appendix A). A slight shift in diffraction peaks to the low-angle side features lattice expansion by increasing Pt/Ni ratios from 0.4 to 1.0. Meanwhile, the suppression of NiO*_x_* peaks again consistently revealed increased shell coverage with Pt loading. An even closer inspection of *D_avg_* values in (111) and (200) facets reveals the morphology of NCs. The Pt-CNT possessed the highest D (111)/D (200) ratio of 1.33 (higher D (111)) and thus the largest extent of preferential crystal growth along the (111) facet among the experimental NCs. Such a result rationalizes the intrinsic nature of the atomic arrangement in close-packed facets in Pt metal. For Ni@Pt-1004, compared to that of Pt-CNT, the D (111)/D (200) ratio decreased by 0.22, which could be attributed to the competition of crystal growth between galvanic replacement of Pt^4+^ to Ni atoms in the open-(High-Miller-index facets) and Pt deposition in close-packed facets (111). In this status, a semi-coherent lattice match was found between facets with truncated surfaces, which can be explained by the formation of twin boundaries as revealed on HRTEM images (Figure 1a). For Ni@Pt-1010, high contents of Pt^4+^ ion reduction and deposition on the NC surface induced severe galvanic replacement to the NiO_*x*_ core. Such a phenomenon mostly occurred at interfaceted corners and edges resulting in a high content of surface truncation in NCs. In this event, NCs tended to form a spherical shape (Figure 1d) due to the strong competition of galvanic replacement to Pt deposition, which was confirmed by a substantially reduced D (111)/D (200) ratio (0.95) compared to that of Pt-CNT.

After adding different contents of Au to Ni@Pt-1004 NCs (Figure 2a), changes in the lattice strain and crystallinity were obvious due to atomic restructuring. In the case of Ni@Pt-1004 NCs, because of a lower Pt-content (Pt/Ni ratio of 0.4), complete coverage of the Ni-core from Pt atoms was not possible. Thus, with a lesser Au content (NPA-1004006), most of the Au atoms were intercalated with Pt atoms, which aggregated between NCs as revealed by the smeared diffraction peaks across the (111) and (200) facets. Restructuring by formation of Au-rich areas at high-order facets (i.e., (200)) between NCs was significant. These characteristics resulted from the spontaneous trans-metalation between Au^3+^ ions and Ni metal atoms (galvanic replacement) accompanied by redeposition of Ni/Au atoms in shell crystals. Such a phenomenon was consistently proven by the suppressed diffraction signals (peaks M1 and M2) of the Au (111) and (200) facets. Further increasing the Au loading to 9 wt.% (NPA-100402) led to obviously increased diffraction signals for the Au (111) and (200) facets, which revealed the presence of discrete Au clusters (~2 nm) on the surface. At the same time, some of the Au atoms tended to deposit on the shell and core regions. Restructuring of Ni@Pt-1010 by interacting with Au^3+^ ions showed a completely different behavior compared to that of Ni@Pt-1004. As shown in Figure 2b, diffraction peaks *X*_1_ and *X*_2_ were not smeared; on the other hand, they were enhanced by the increasing Au contents from 2.0 to 9.0 wt.%. Such characteristics reveal that the long-range ordering of Ni@Pt NCs was improved by Au decoration. In the absence of diffraction peaks from the Au metal phase and the enhanced Au-4f photoemission peaks (Appendix A), one can notice that the galvanic replacement of Au^3+^ to the NC surface was suppressed, which suggests conformal deposition of Au atoms in the Pt-rich shell.

The XPS analysis was performed in order to investigate the surface chemical composition (1~2 nm from the surface) and binding energy (BE) of elements in experimental NCs. The incident X-ray with an excitation energy of 650 eV corresponding to a probing depth of ~2.6 nm was employed to probe the Pt-4f, Ni-2p, and Au-4f orbitals. Figure 3 reveals the typical fitted XPS spectra in the Pt-4f region of experimental NCs. In the Pt-4f spectrum, doublet peaks at 71 and 74 eV, respectively, emerged as photoelectron emission lines from the Pt-4f_7/2_ and Pt-4f_5/2_ orbitals. The peaks are further deconvoluted to separate the signals from different oxidation states, and corresponding results are given in Table 2. Through an analysis of XPS patterns of Ni@Pt-1004 and Ni@Pt-1010 NCs (Figure 3a,b), it can be seen that most of the Pt was in a zero-valence state (metallic state). In contrast, from the XPS spectra of Ni-2p orbitals (Appendix A), it is clearly evident that Ni is present in an oxidized (NiO_*x*_) form in both Ni@Pt NCs (i.e., Ni@Pt-1004 and Ni@Pt-1010). An even closer inspection of the intensities of the XPS spectra of Ni-2p (Appendix A) revealed that NiO_*x*_ signals in Ni@Pt-1010 NCs were very much suppressed compared to those of Ni@Pt-1004 NCs. These spectral characteristics confirmed the core-shell structure of Ni@Pt-1010 NCs comprising a Ni core and Pt in the outermost layer. Meanwhile, the profound intensities of Ni-2p emission peaks in Ni@Pt-1004 NC refer to an incomplete Pt-shell over an underlying Ni-crystal. Those results again confirm prior HRTEM and XRD findings.

For Au-cluster-decorated Ni@Pt-1004 NCs (Figure 3a), significant restructuring of the surface atomic arrangement was observed. NPA-1004006 NCs (2 wt.% Au) exhibited a higher extent of zero-valent Pt (Pt^0^) contents on the surface compared to that of NPA-100402 (9 wt.% Au). More evidence of such atomic restructuring came from the XPS analysis of Ni-2p orbitals (Appendix A). Compared to that of Ni@Pt-1004, we observed that the intensity of the Ni-2p emission peak was gradually suppressed with an increasing amount of Au of 2 to 9 wt.%. Intensities of emission peaks in an XPS spectrum are positively related to the electron density in probing orbitals of target atoms. Therefore, the higher intensity found in the curve of NPA-1004006 revealed that its most abundant 2p electrons were Ni atoms, compared to that of NPA-100402 NCs. Similar spectral changes were observed for Au-cluster-decorated Ni@Pt-1010 NCs (Figure 3b). Obtained XPS results were very consistent with former structural characterizations. The difference in binding energy of the 4f_7/2_ orbital of zero-valent Pt (Pt^0^) was not obvious in the experimental NCs, revealing that electron relocation between Pt and neighboring atoms was nearly absent. For comparison XPS spectra of Ni@Pt-1010, NPA-1010006 and NPA-101002 at Ni-2p orbitals are compared in Appendix A.

XPS spectra of experimental samples of Au-4f orbitals are compared in Appendix A. Accordingly, intensities of Au-4f_5/2_ and Au-4f_7/2_ peaks increased with Au contents from 2 (NPA-1004006) to 9 wt.% (NPA-100402). Such a phenomenon shows the increasing exposure of Au with loading, which again consistently proves the formation of Au clusters in Ni@Pt NCs with an incomplete shell structure. In the case of Ni@Pt-1010, when the Au loading was 2.0 wt.%, the doublet peaks in the Au-4f orbital were insignificant. Such a result can be complimentarily explained by the crystal structure parameters. As indicated in Figure 2b, a significant improvement in Pt crystallinity was found, and diffraction peak shifts were absent when decorated with 2 wt.% of Au atoms on the Ni@Pt surface. These features indicate the formation of atomic Au clusters on the NiO_*x*_@Pt surface. Therefore, the presence of weak emission peaks suggests a discrete and smeared 4f orbital of Au atomic clusters that were finely dispersed in surface defect sites of NiO_*x*_@Pt NCs. By increasing Au to 9 wt.%, pronounced Au-4f peaks rationalized the formation of Au sub-nano- or nanoclusters, as revealed by presence of a diffraction shoulder on the low-angle side of *X*1 (111) and a pronounced intensity in *X*2 (200) peaks.

By cross-referencing results of the physical characterization, the effects of Au^3+^ loading and Pt contents on the evolution of atomic structures of Ni@Pt NCs was systematically determined, and corresponding structural models are given in Scheme 1—where the upper and lower layers respectively present changes in the atomic structure with increasing Au^3+^ loading for Ni@Pt-1004 and Ni@Pt-1010. Accordingly, a significant galvanic replacement on oxidation followed by dissolution of Ni^0^ to Ni^2+^ appeared by interacting Ni@Pt-1004 with 2 wt.% of Au^3+^ (step i in the upper layer of Scheme 1) with the reaction of Au^3+^ + Ni^0^ (or Pt^0^) Au^0^ + 3/2Ni^2+^ (or 4/3Pt^4+^), where Au^3+^ has a higher selectivity for Ni^0^ than Pt^0^ due to the larger electronegativity difference. In this event, Au atoms tended to penetrate the core region, thus resulting in the coexistence of nanosized Au and Pt clusters in NiO_*x*_@Pt NPs. By increasing the loading to 9 wt.%, galvanic replacement between Au^3+^ ion and core crystals was further enhanced, which caused the severe interparticle agglomeration by dissolution of core metal atoms accompanied by the rapid redeposition of residual metal ions between interfaces (i.e., regions a, b, and c in step ii of the upper layer of Scheme 1) of the NPs. Compared to those of Ni@Pt-1004, the effects of Au^3+^ loading on the structural evolution were suppressed by the high contents of the Pt shell structure in Ni@Pt-1010. As shown in the bottom layer of Scheme 1, Au^3+^ tended to be adsorbed and was reduced by NaBH_4_ to form atomic clusters on the NP surface with a loading of 2.0 wt.% (NPA-1010006). By increasing the loading to 9.0 wt.%, the Au^3+^ ions tended to form homoatomic bonds and thus grew into sub-nanometer crystals on the NPA-101002 surface (step ii in the bottom layer of Scheme 1). These atomic structural arrangements provide direct information explaining ORR activities of the experimental NCs.

In heterogeneous catalysts, dissociation of chemisorbed oxygen molecules (i.e., the oxygen adsorption strength) is a cardinal performance-determining factor in ORRs. Lowering the oxygen adsorption energy reduces the applied energy for initiating ORRs at reaction sites and relocating them to neighboring atoms. In this way, the reaction kinetics and ORR activities of NCs can be substantially improved. In this study, the surface composition design of catalysts within the sub-nano scale played a key role in ORR performances of the NCs. By cross-referencing physical inspection results (the upper layer of Scheme 1), the surface chemical configuration of NiO_*x*_@Pt comprised mixtures of sub-nano Au and Pt clusters in the shell region and Au clusters intercalated with NiO_*x*_ in the core when the Pt/Ni ratio was 0.4 and Au was 2.0 wt.% With an Au content of 9.0 wt.%, severe interparticle agglomeration due to galvanic replacement (Au^3+^ + Pt/Ni → Au + Pt^4+^/Ni^2+^) accompanied by Au crystal growth and redeposition of residual metal ions between NCs occurred. All three pathways dramatically reduced the degree of heteroatomic intermixing on the surface among reaction sites; therefore, ORR activities of those NCs were substantially suppressed.

Results of the electrochemical analyses consistently elucidated the above scenarios. Figure 4a compares CV curves of the commercial Johnson Matthey-Pt/C catalyst (Johnson Matthey-Pt/C) with the experimental NCs (i.e., Ni@Pt-1004, NPA-1004006, and NPA-100402). Electrochemical active surface areas (ECSAs) are calculated based on corresponding CV curves using the oxygen desorption peak in the backward potential sweeping curve (detailed ECSA data of various ORR catalysts are listed in Appendix A). Three distinctive potential regions are found in a CV curve, including an under-potential deposition of hydrogen (UPD-H) region at 0 < E < 0.4 V, a double-layer region between 0.4 and 0.6 V, and chemisorption of oxygen species at >0.6 V vs. the RHE because of hydrogen adsorption/desorption, OH- ligand chemisorption, and the formation of alpha Pt oxide (E_O_^ads^; forward scan) as well as a reduction in Pt oxides (E_O_^des^; backward scan). In this way, the position and width of each peak are susceptible to the chemical composition and structure of the NC surface. For the Johnson Matthey-Pt/C, positions of two characteristic peaks (H_1_ and H_2_) in the forward scan denoted the potential to be applied for dissociation of H^+^ from close-packed (111) and opened (200) facets and the corresponding current, respectively. In contrast, peaks H_1_* and H_2_* respectively refer to current responses of H^+^ adsorption in the (111) and (200) facets. For Ni@Pt-1004, compared to the CV profile of the Johnson Matthey-Pt/C, a downshift of peaks H_1_ and H_2_ in the forward scan and an upshift of peaks H_1_* and H_2_* in the backward scan refer to a decreased energy barrier for redox desorption/adsorption of H^+^. As consistently shown by XRD observations, a substantially higher intensity of peak H_1_ than that of peak H_2_ (i.e., weakened H^+^ interactions on (200) facets) revealed preferential crystal growth at (111) facets in all experimental NCs.

Compared to that of the Johnson Matthey-Pt/C, Ni@Pt-1004 showed a higher surface area for H_2_ evolution as revealed by the larger area of the UPD_H region. Meanwhile, the broadened and smeared UPD_H peaks revealed a high density of surface defects in Ni@Pt-1004. This statement is consistently illustrated by the pronounced oxygen adsorption peak (E_O_^ads^) with a potential shift by ca. ~0.17 V (i.e., easy oxidation of the Ni@Pt-1004 surface) compared to that of the Johnson Matthey-Pt/C. Compared to the CV profile of NCs without Au decoration, a slight amount of Au decoration reduced the surface defect sites of Ni@Pt-1004, as consistently revealed by the significant suppression of the E_O_^ads^ and E_O_^des^ peaks in NPA-1004006. Moreover, the position of the oxide reduction peak (E_O_^des^) was upshifted to high-potential sites at the same time. These two observations integrally bring out the fact that atomic decoration by Au clusters can fix the defect and simultaneously suppress the surface oxidation on the Ni@Pt-1004 surface. In the presence of discrete 4f orbitals in atomic clusters, a strong repulsive force to the chemisorbed O (O^ads^) was formed at Au atoms. This relocated the O^ads^ to neighboring atoms around the Pt and Au interfaces and thus substantially boosted the ORR kinetic current (*J*_k_) of NPA-1004006 to ∼75 mA cm^−2^ (details discussed below in the LSV analysis, Figure 4c). For NPA-100402, the H^+^ adsorption peak, “H_1_*”, in the backward scan together with a smeared CV profile in forward scan was observed in the UPD_H region. Such a feature can be rationalized by its nanostructure, where the surface of the NC consists of nanosized Au/Pt clusters. Meanwhile, a severe interparticle agglomeration by the strong galvanic replacement accompanied by rapid reduction of residual metal ions was found (Figure 1c, Scheme 1); therefore, the heteroatomic intermix and the amount of reaction sites dramatically decreased. Formation of cluster-in-cluster structures turned 4f orbitals from a discrete state into a band structure. In this state, both the Au and Pt atoms possessed bulk properties, and consequently the *J*_k_ of NPA-100402 was dramatically reduced by 87% (65.62 mA cm^−2^) to 9.4 mA cm^−2^ (Appendix A), compared to that of NPA-1004006. This value was even lower than that of Ni@Pt-1004, indicating that the impact of the heterogeneous interface between nanoclusters on ORR activity was limited.

On the other hand, as shown in Figure 4b, changes in profiles in CV curves were insignificant by decoration with Au atoms on Ni@Pt-1010. In the case of 2.0 wt.% Au decoration, a slight suppression of both E_O_^ads^ and E_O_^des^ was found for NPA-1010006 NCs compared to that of Ni@Pt-1010. These characteristics resemble the same redox response to O adsorption and desorption when decorating Au atomic clusters on the Ni@Pt-1004 surface. Increasing the Au content to 9.0 wt.% did not further suppress the redox peaks of O evolutions (E_O_^ads^/E_O_^des^); however, they moved in an opposite direction. Due to the strong preference for homoatomic bonding, Au atoms tended to form sub-nano clusters instead of a conformal coating or defect sites on NC Pt surfaces. Those characteristics reduced the heteroatomic intermix and amount of O_2_-splitting sites in NCs. The former suppresses intrinsic activity, and the latter reduces the number of active sites; therefore, the *J*_k_ of NPA-101002 was substantially reduced by 70% (to ∼22 mA cm^−2^), compared to that of NPA1010006. Again, given that most of the Au atoms were deposited as sub-nanometer clusters or atomic clusters (shown by absence of an Au diffraction peak in the XRD pattern of Figure 2b), a high *J*_k_ (∼22 mA cm^−2^) was expected.

To further rationalize the impacts of Au decoration and Pt contents on ORR activities of Ni@Pt NPs, an LSV analysis was employed. Figure 4c,d demonstrates LSV curves of NPA compared to control samples (Ni@Pt-1004 and Ni@Pt-1010) and the Johnson Matthey-Pt/C, where corresponding electrochemical parameters are summarized in Figure 5, and Appendix A. As shown in Figure 4c and Appendix A, the onset potential (Voc vs. the RHE) of experimental samples followed the trend of the Johnson Matthey-Pt/C (0.910 V) < NPA-1004006 (0.964 V) < NPA-100402 (0.969 V) < Ni@Pt-1004 (0.990 V). Among them, the highest Voc indicates the lowest activation energy for initiating ORRs on Ni@Pt-1004 surfaces and can be rationalized by a large extent of heteroatomic intermix between Pt and Ni sub-nanoclusters. On such a surface, Pt clusters are in charge of splitting (dissociating) the oxygen molecule (O_2_). After O_2_ splitting, the two chemisorbed oxygen atoms are relocated to the high-oxygen-affinity Ni atoms for the subsequent reduction reaction. Since the dimensions of Pt clusters are quite small (i.e., the interface ratio between Ni domain is high), a rapid relocation of chemisorbed oxygen atoms to Ni sites is expected; therefore, the *J_k_* of Ni@Pt-1004 doubled compared to that of the Johnson Matthey-Pt/C. By decorating with 2.0 wt.% Au atoms on Ni@Pt-1004, the atomic Au clusters reduced surface defects of the Pt shell and covered the Ni core crystal. In this event, activation energy for O_2_ splitting increased, which reduced the intrinsic activities of surface sites and thus the Voc by 0.026 V. Such a phenomenon would seemingly suppress the activity of surface sites on the NC surface; however, it actually went the opposite way. As revealed by results of physical structural inspections, decoration with a slight amount of Au atoms (2.0 wt.%) resulted in a surface structure comprising combinations of Au, Pt, and Ni sub-nanoclusters on the NC surface. The presence of sub-nano or atomic Au clusters in defect sites protected the NC surface from oxidation (shown by the suppressed E_O_^ads^ peak in the CV curve). In this event, relocation of the O^ads^ was dramatically facilitated, resulting in a quantum leap of *J_k_* by 6.7-fold compared to that of NCs without Au decoration (i.e., Ni@Pt-1004). By increasing Au to 9.0 wt.%, compared to that of NPA-1004006, the *J_k_* of NPA-100402 substantially decreased by ~87% to 9.4 mAcm^−2^. This value was almost the same as that of Ni@Pt-1004, indicating that all reaction sites (Pt, Au, and Ni) possessed metallic properties of NPs (bulk) without facilitation of heteroatomic intermixing or ligand effects in ORRs. Meanwhile, as shown in Figure 4c, similar slopes of diffusion (V < 0.8 V vs. the RHE) and kinetic limit (V > 0.8 V vs. the RHE) current regions indicated that the redox responses of Ni@Pt-1004 were not greatly influenced by a high Au loading. Such electrochemical properties are understandable due to the fact that Au atoms tend to form large nanocrystals by homogeneous crystal growth on the surface and galvanic replacement in the core region of Ni@Pt NCs. Compared to those of Ni@Pt-1004, Au decoration showed similar effects on ORR activity of Ni@Pt NCs with a conformal Pt shell (Ni@Pt-1010). Figure 4d compares LSV curves of experimental NCs (NPA-1010006 and NPA-101002) with those of the control sample (Ni@Pt-1010) and the Johnson Matthey-Pt/C; corresponding electrochemical properties are summarized in Appendix A. Accordingly, VOC and E_1/2_ followed the same trend as that of Ni@Pt-1004 with increasing Au contents from 2.0 to 9.0 wt.%. The same scenario to NPA NCs with low Pt contents held for NPA-1010006, except that Au atoms did not penetrate into the core region to form nanosized clusters, therefore NPA-101002 retained a high *J_k_* value (21.99 mA cm^−2^) in ORRs.

Surface activities (SAs) of electrocatalysts in ORRs are calculated by normalizing the *J_k_* to the ECSA of the oxygen desorption region in the CV curve. These values are an important index for the average intrinsic activity of reaction sites on NC surfaces. As shown in Appendix A, the ECSA was 81.2 cm^2^ mg^−1^ for Ni@Pt-1004 and decreased to 50.0 cm^2^ mg^−1^ by decorating with 2.0 wt.% Au atoms, again proving that Au atoms tended to reside in defect sites of the Pt shell. Further increasing the Au content to 9.0 wt.% did not affect the ECSA value. This result, consistent with that proven by the XRD analysis, suggests that Au atoms tended to grow in homoatomic nanocrystals instead of capping on the Pt shell surface. Accordingly, SAs were 13.89 mA cm^−2^ for NPA-1004006 and 1.83 mA cm^−2^ for NPA-100402. With similar ESCAs, the significantly enhanced SA elucidates conformation of the substantially improved intrinsic ORR activity by the presence of atomic/sub-nano Au clusters simultaneously with Pt and Ni nanoclusters on the NPA-1004006 surface. For Ni@Pt with a conformal Pt shell, Au decoration mainly occurred on the NPA-1010006 surface, as indicated by a reduction in the ECSA (64.4 cm^2^ mg^−1^) by 11.9% compared to that of Ni@Pt-1010 (73.1 cm^2^ mg^−1^). Compared to that of NPA-1010006, the ESCA increased by 11.3% when decorated with 9.0 wt.% of Au on the Ni@Pt-1010 surface, which can be attributed to the formation of sub-nano Au clusters on the NC surface. In this event, SAs were 5.64 mA cm^−2^ for NPA-1010006 and 1.61 mA cm^−2^ for NPA-101002. Compared to those of NPA with Pt/Ni = 0.4, the substantially suppressed SA reveals the truth that the combination of atomic Au/Pt clusters with Ni atoms in neighboring sites can support exceptional reaction kinetics of ORRs (i.e., *J_k_* and SA).

Mass activity (MA) refers to the current density per unit weight of active sites and is calculated by normalizing the residual current at 0.85 V vs. the RHE with respect to the loading amount of metal Pt in NCs. As illustrated in Figure 5a, the MA of Ni@Pt was slightly improved by 26% compared that of the Johnson Matthey-Pt/C. By adding 2.0 wt.% Au atoms on the surface, the MA of NPA-1004006 substantially improved by 7.1-fold compared to that of Ni@Pt-1004. With a slight increment of noble metal loading, such a dramatic improvement in the MA depicts the truth for boosting the activity of NCs by syngeneic collaboration between sub-nano Au, Pt, and Ni domains in the reaction pathways in ORRs. Such a scenario was further confirmed by the MA of NPA-100402. In this NC, the MA was significantly reduced by 87% compared to that of NPA-1004006. Given that the difference in noble metal loading was small (7.0 wt.%), the dramatic difference in the MA again proves changes in intrinsic activities instead of mass differences. The same phenomenon exists in changes of the MA with respect to Au loading in Ni@Pt-1010 again complementarily proves the synergetic effects on ORR activities of NCs. Electrochemical results of control samples with commercial catalyst has been compared in Appendix A and corresponding parameters has been summarized in Appendix A.

## 4. Conclusions

CNT-supported NCs with a Ni/NiO_*x*_ base and an Au cluster-modified Pt-shell were synthesized via self-aligned wet-chemical processes with variable shell thicknesses (Pt/Ni ratios of 0.4 and 1.0) decorated with different contents of Au atoms (2 and 9 wt.%). Results of physical structural characterizations combined with electrochemical analyses proved that surface coverage of the Pt-shell along with depth and distribution of Au clusters significantly affected inner structural configurations and thus the ORR activities of bimetallic Ni@Pt NCs. For Ni@Pt-1004 NCs, because a lower Pt-content and lower surface coverage were adopted, Au atoms tended to form sub-nanoclusters accompanied by Pt and Ni on the NC surface at a loading of 2.0 wt.% Such an NC exhibited the highest *J_k_* (75.02 mA cm^−2^), SA (13.89 mA cm^−2^), and MA (694.49 mA mg_Pt_^−1^) among the experimental NCs due to the synergetic collaboration between oxygen-inert and -affinity sites on the surface. When increasing the loading to 9.0 wt.%, Au atoms tended to penetrate into the core region and grow into homoatomic clusters on the NC surface. Both characteristics reduced the heteroatomic intermix and surface ratio, and therefore, turned the redox properties of NCs into a bulk nanocrystal state. For the case of Ni@Pt with a conformal shell, Au atoms tended to form atomic clusters on the NC surface which exhibited a *J_k_* of 73.78 mA cm^−2^ corresponding to an MA of 362.9 mA mg_Pt_^−1^ and SA of 5.64 mA cm^−2^. Compared to those with low Pt contents, the lower electrochemical performances of NPA with a Pt/Ni ratio of 1.0 consistently explained the local syngeneic effects on the NC surface. In brief, robust methods to synthesize bimetallic NCs with different extents of surface decoration were developed in this study. We demonstrated that such processes can be adopted to control the identity and local structural disorder on NC surfaces. By proper control of the surface decoration loading, the ORR performance of NiO@Pt catalysts was improved by 7.1-fold in the optimal case. These results elucidate a new prospect of heterogeneous catalyst design. It realizes a compact co-catalyst with different sub-nanometer components collaborating together to share intermediate steps in redox reactions and thus enabling facilitation of the activity of electrocatalysts.

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
