# Peer review of "Conformational Effects of Pt-Shells on Nanostructures and Corresponding Oxygen Reduction Reaction Activity of Au-Cluster-Decorated NiOx@Pt Nanocatalysts"

_nanomaterials, 2019, doi:10.3390/nano9071003_

Round 1
Reviewer 1 Report
Dear Authors,
Thank you for your research and for the good work has been performed and described. It was interesting to read the manuscript indeed.
At the same time, let me put a few remarks regarding the text as a reviewer.
First of all, please, do pay much more attention to the text formatting. It was difficult to read due to a large number of not well-written formulas etc. I have marked just at the beginning the places should be corrected, but you'll need to check all the text until the end. For a better impression, please, do not send a text before this sort of very basic formatting will be completed.
There are several scientific questions should be discussed as well.
1) The accuracy of the measurements has a strong influence on the data format you should insert into the text and tables.
a) I've doubts regarding the possibility to get 17.16 or 10.23 folds values (lines 26 & 29 correspondingly) as accurate as you have shown! It's better to write "nearly 17 and 10" instead.
b) Another example of not proper data format (from my point of view): lines 199-200. It is necessary to submit an estimation: how accurate you can calculate the size of NPs via Scherrer's equation? Is there the accuracy as high as ±0.02 nm?
c) line 213 contains 40.24 & 46.41 (2θ, degrees) values, could you specify, please, what would be a suitable range of the numbers in order to be not quite confused by the values have been shown? I'd offer to write 40.2 & 46.4 (2θ, degrees). But if you have another opinion, please, explain the data quality via a standard's powder XRD pattern, etc.
d) lines 383 & 405 etc, the numbers 75.02 & 21.99 mA●cm-2 initiate the same question, it is necessary to determine and to show errors bars for this method as well.
Please, do check all the data around the article and try to write the numbers correctly.
I hope these comments will help you to improve the manuscript.
Best wishes!

Author Response
The authors appreciate to the kind reminder for correcting the contents and formats. The manuscript has been corrected accordingly.

Reviewer 2 Report
Bhalothia et all report the use of several ternary metallic nanocatalysts (NCs) on carbon nanotube (CNT) support for application in the oxygen reduction reaction (ORR). Overall the theme is quite interesting.
The manuscript itself requires extensive revision, namely in the scientific presentation, where no attention was given to the writing of molecules and units.
The characterization is incomplete, since a great care was given to the surface of the catalysts but considering the this is a composite with several interfaces, an EDX should be performed and a FEG-SEM also to confirm scheme 1.
A brief characterization (HRTEM, EDX and XPS) and discussion of the best nanocatalyst, after reaction, should be presented, so that some of the claims made by the authors can be confirmed.
Author Response
The authors appreciate the kind help on correcting several important contents.
The manuscript has been corrected according to the comments. Meanwhile, additional information for the comment has been provided.

Round 2
Reviewer 1 Report
Dear Authors, at present the manuscript looks almost fine, a couple of technical errors at the lines 391 and 450 have to be corrected.
I hope the article will be published shortly. Best wishes!

Author Response
The authors appreciate the kind reminder for correction of the technical issues in lines 391 and 450. They have been corrected to the proper format accordingly.
Reviewer 2 Report
The authors replied the questions.
Author Response
The authors appreciate the acknowledge of our efforts.
We believe that the contents of this manuscript do contribute to the new catalyst design regimes.